# Hospitalized Patients with Medically Unexplained Physical Symptoms: Clinical Context and Economic Costs of Healthcare Management

**DOI:** 10.3390/bs9070080

**Published:** 2019-07-19

**Authors:** Nicola Poloni, Ivano Caselli, Marta Ielmini, Michele Mattia, Alessandra De Leo, Marco Di Sarno, Celeste Isella, Alessandro Bellini, Camilla Callegari

**Affiliations:** 1Department of Medicine and Surgery, Division of Psychiatry, University of Insubria, Viale Luigi Borri 57, 21100 Varese (VA), Italy; 2Family Therapy Center, Via San Salvatore 7, 6902 Lugano, Switzerland; 3Department of Psychology, University of Milan–Bicocca, Piazza dell’Ateneo Nuovo 1, 20126 Milan (MI), Italy

**Keywords:** Health Care Costs, Community Psychiatry, Medically Unexplained Physical Symptoms, Psychosomatic Medicine, Somatic Symptoms Disorder

## Abstract

Medically Unexplained Physical Symptoms (MUPS) are physical symptoms without a medical explanation. This study collected data from hospitalized patients presenting MUPS, aiming to draw a clinical and socio-demographic profile of patients with MUPS, to explore psychopathological correlations of Somatic Symptoms Disorder (SSD) diagnosis, and to estimate economic costs related to hospital management for MUPS. The cross-sectional study consisted in the evaluation of data referring to hospitalized patients admitted between 2008 and 2018 in a teaching hospital in Northern Italy. A total of 273 patients presenting MUPS have been hospitalized. The sample showed a prevalence of female, married and employed patients. The most frequent wards involved are Neurology, Internal Medicine and Short Unit Stay. The most common symptoms found are headache, pain, syncope and vertigo. There is no evidence that a history of medical disease is associated with a diagnosis of SSD. A personality disorder diagnosis in patients with MUPS was associated with increased probability of having a diagnosis of SSD. A marginally significant positive association emerged with anxiety disorders, but not with depressive disorder. The overall estimated cost of hospitalization for patients with MUPS is 475′409.73 €. The study provides the investigation of a large number of patients with MUPS and a financial estimate of related hospitalization costs.

## 1. Introduction

Medically unexplained physical symptoms (MUPS) are physical symptoms without a medical explanation. This definition is used to imply somatic symptoms that cannot or have not been sufficiently explained by organic cause after a thorough physical, laboratory and instrumental examination [1]. The persistence of distressing physical symptoms is linked to a huge individual and societal burden and unmet clinical need [2]. MUPS are related with high levels of psychological distress and can lead to an important functional impairment, interfering with work productivity and daily functioning. An association with a high utilization of the healthcare resources and elevated costs are shown in the professional literature [3].

In the official classifications of the Diagnostic and Statistical Manual of Mental Disorder IV-text revision (DSM IV-TR) and International Classification of Diseases 10^th^ revision (ICD-10), the presence of medically unexplained symptoms was a criterion to fulfill the diagnosis of somatoform disorder. This diagnosis was introduced for the first time in DSM-III [4] and in ICD-10 [5], to try to create a new group that was useful to collect all physical symptoms in which no organic cause was demonstrable. 

In DSM-5 [6], the nature of the physical symptoms is no longer a criterion for somatoform disorders. In fact, DSM-5 focuses on the way a patient emotionally, cognitively and behaviorally copes with the physical symptoms. According to the Somatic Symptoms Disorder (SSD) classification, even if a patient is suffering from chronic medical conditions, they can also be diagnosed with SSD and receive treatment [2]. The previous classifications were considered difficult to use in clinical practice, especially among general practitioners and non-specialists, because of their rigid categories [7]. On the other hand, in DSM-5, the somatic symptom and related disorders chapter has a limited clinical utility and presents some ambiguity [8,9,10]. This diagnostic classification reduces the importance of medically unexplained symptoms and emphasizes the psychological criteria and the functional impairment experimented by the patient.

Furthermore, in epidemiological studies, those which were based on DSM criteria for somatoform disorder resulted in low prevalence of this disease, differently from what we observe in the clinical practice [11,12]. In the opinion of many authors, this gap is due to the fact that the diagnostic criteria do not correspond to reality [13]. In 2004 [14], a systematic review of all epidemiologic studies collected 47 papers in the general population and general medicine. It is interesting to note that using standard criteria for somatization disorder, the mean prevalence was 0.4% in the general population and using reduced criteria, such as Somatic Symptom Index (SSI) [11], the results ranged from 4.4% to 19%. It is also interesting to note that in the prevalence studies there is a wide range of prevalence, which often depends on the sample analyzed, for example, in a Dutch study published in 2004, the prevalence of somatoform disorders in general practice was 16.1% [15].

If MUPS are considered not as a feature of a specific disorder but as a health problem itself, a high prevalence of these problems can be noted. Up to one-third of all people presenting with physical symptoms have MUPS [16], but also within the MUPS category, the studies showed wide heterogeneity in terms of the prevalence rates [17]. MUPS are frequently associated with the female gender [18,19] and low socio-economic status [20]. The mean age in which MUPS are more frequent varies between different studies [21]. MUPS are often associated with psychiatric disorders, with a considerable degree of diagnostic overlap with depression, anxiety and panic disorder and substance abuse [3], nevertheless these patients are seen by a psychiatrist very late in their history of disease. MUPS are the most commonly found symptoms in primary care and they often occur even in organic pathology [3,22]. They also have a high prevalence across secondary care settings and they are responsible for a huge proportion of disability and decreased quality of life among the general population [23].

These patients represent an important clinical phenomenon with considerable direct and indirect economic consequences. In the USA and in the UK, several studies have attempted to calculate either the aggregate or individual cost of conditions associated with somatization, highlighting different estimates [23,24]. Previous studies on somatic symptoms disorder support the evidence for an unfavorable outcome of conditions involving persistent functional somatic symptoms, but these studies are mainly based on self-report questionnaires and/or less well-defined diagnostic constructs [25].

As far as we know, there are few studies on medically unexplained symptoms in patients admitted to hospital in the scientific literature. Moreover, correlations of somatic symptoms and associations with clinical variables are often unclear and must be discussed. Thus, the present study provided for the collection of data from hospitalized patients presenting medically unexplained physical symptoms (MUPS) referring to different hospital wards, aiming at the following outcomes: (1) to draw a clinical and socio-demographic profile of hospitalized patients with MUPS; (2) to explore psychopathological correlations of SSD diagnosis; (3) to estimate economic costs related to healthcare utilization of MUPS.

## 2. Materials and Methods

The cross-sectional study consisted of the evaluation of data referring to all hospitalized patients admitted between 2008 and 2018 in the wards of a teaching hospital in Northern Italy (Deliberate n. VIII/4221, 28 February 2007).

The research involved the Internal Medicine, Neurology, Infectious Disease, Orthopedics, Otorhinolaryngology and Emergency wards; Short Stay Unit data were available from 2014, Emergency and Transplant Surgery data from 2015 and Psychiatry data from 2012. Data from the Short Stay Unit and Emergency and Transplant Surgery were available from the year these wards were opened. Data from the Psychiatry ward were computerized from 2012. Emergency ward data collected referred to the period from November 2017 to November 2018. All data were recruited between January 2018 and January 2019.

Hospital discharge letters were analyzed by three psychiatry section clinicians from the hospital software. The clinicians were not directly involved in analyzed patients’ diagnosis and treatment.

Data from patients fulfilled the following inclusion criteria: age > 18; be an inpatient in the teaching hospital; present symptoms with apparently no medical cause, or whose cause remains unclear (Medically Unexplained Physical Symptoms); have a diagnosis of *Somatoform Disorder* or *Somatic Symptoms Disorder and related disorders* by non-specialists (according to DSM-IV-TR and DSM-5; since Italian statistical medical recording is ICD, diagnoses have been made through the ICD code conversion Table); present all test clear. No excluding criteria were used.

The following socio-demographic and clinical variables were evaluated: gender, age, marital status, employment, diagnosis or diagnostic hypothesis in admission and discharge, personal medical history, presence of previous or concurrent psychiatric comorbidities, length of hospitalization, healthcare costs, medical examinations, psychiatric evaluation, pharmacological treatment.

The economic costs of each hospitalization were obtained from the economic value sheet combined with the discharge letter uploaded on the electronic register of the hospital. When unavailable, the average costs of hospitalization for each patient were estimated by the Management control division of the hospital. The costs of laboratory and instrumental examinations were found on the document “Nomenclature tariff of the specialist outcare patient” (Ministerial Decree 216, 12 January 2017) DPCM 2017) of the Italian National Health System. 

All patients provided a general written informed consent to the processing of personal data as part of the routine quality check processes.

Patients’ data were made anonymous, obscuring sensitive information used in the research to protect the recognizability of the patients, according to the Italian legislation (D.L. 196/2003, art. 110—24 July 2008, art. 13). 

The Provincial Health Ethical Review Board (Ethics Committee of Insubrias—Varese, Italy) was consulted prior to the beginning of the study; it confirmed that, as the research was a cross-sectional retrospective study, it did not need authorization from the Board. 

The study was carried out in accordance with the ethical principles of the Declaration of Helsinki (with amendments) and *Good Clinical Practice*.

To summarize epidemiological and clinical characteristics, descriptive statistics (which include means, standard deviation and demographic variables percentages) were computed.

To better detect the clinical and socio-demographic characteristics of the patients, hospital wards were grouped into different macro-areas: Medical wards: Internal Medicine, Neurology, Infectious Disease, Short Stay Unit; Surgical Wards: Emergency and Transplant Surgery, Orthopedics; Emergency Ward; Psychiatry; Otorhinolaryngology.

Statistical analyses were performed on data from medical specialties, including surgical wards, Psychiatry and Audiovestibology. Emergency ward data were not computed because of the lack of patients’ personal information. 

Analyses were conducted to investigate specific issues regarding the probability of having a diagnosis of somatic symptoms disorder in our sample of patients with MUPS. In particular, chi-square tests (*χ*^2^) were used to investigate whether there were differences in the distribution of the diagnosis of somatic symptoms disorder in the two genders, as well as in the diverse conditions of civil status and employment. Two multiple logistic regression models were used to evaluate whether a series of medical and psychiatric conditions were associated with an increased probability of having a somatic symptoms disorder diagnosis. In particular, we tested a model with medical diseases as independent variables (including previous medical history, neurological anamnesis, fibromyalgia, neoplasms, metabolic diseases, autoimmune diseases, endocrinological diseases, infective diseases, medical diseases, surgery, and accidents), and a second model with psychiatric disorders as independent variables (Depressive Disorder, Anxiety Disorder, Personality Disorder). In both models, all independent variables were dichotomic categorical variables, with a value of 0 indicating no pathology in anamnesis, and a value of 1 indicating the presence of pathology.

All analyses were conducted through the software IBM® SPSS® Statistics version 25.0 (IBM Corp., Armonk, NY, USA) was used [26].

## 3. Results

### 3.1. Socio-Demographics and Clinics

Socio-demographics and clinical characteristics of the sample are showed in Table 1.

The overall number of hospitalizations that were detected was 306. We calculated the total number of patients with MUPS considering that three patients had more hospitalizations in the research period. The distribution of patients in different wards is shown in Table 2. The prevalence of patients with MUPS is shown in the same table, considering the percentage of people hospitalized more than once was under 10%.

The average length of hospitalization in different wards was the following: Medical Wards (7 days); Surgical Wards (5 days); Psychiatry (8 days); Audiovestibology (7 days). 

As shown in Table 3, 46% of the sample (n = 126) patients present no psychopathological comorbidities, of which 65.8% (n = 83) are women and 34.1% (n = 43) are men. 

In previous medical history, 46% of patients (n = 101) patients presented at least one psychiatric disorder in comorbidities, of which 77.2% (n = 78) were women, and 22.7% (n = 23) were men. The most frequently detected diagnosis was: (1) Anxiety Disorder (50%); (2) Depressive Disorder (15%); (3) Somatoform Disorder (3%); (4) Substance Abuse (3%).

Upon discharge, 16.8% of patients (n = 46), of which 65.2% (n = 30) women and 34.7% (n = 16) men, were newly diagnosed with a psychiatric disorder. The most frequent diagnosis was: (1) Depressive Disorder (37%); (2) Anxiety Disorder (35%); (3) Somatoform Disorder (15.5%).

The diagnosis of somatoform disorder was formulated in 7.9% of cases, in 5% of cases the diagnosis was in comorbidity with other psychiatric disorders; in 2.9% of cases without comorbidities.

A psychiatric consultation was requested in 75 admissions and a psychopharmacological treatment was set in 157 cases; in 52 cases, the therapy was prescribed by a psychiatrist. Not including the number of hospitalizations in psychiatry, 138 (50.5%) patients did not receive any psychiatric treatment. The pharmacological treatment consisted of benzodiazepines (10.5%) and Selective Serotonin Reuptake Inhibitors (9.5%), in 30.4%, the treatment consisted of combinations of different classes of drugs. 

Considering the overall hospitalization, the most common symptomatology found for patients with MUPS are: headache (21.9%); pain (14%); syncope (8.8%); vertigo (4.6%). Symptoms per unit are shown in Table 4. 

A total of 6291 admissions to the Emergency Ward were observed in patients with MUPS; this sample is composed by 5735 subjects, 55% of the sample are women (n = 3142), 45% are men (n = 2590). The average age of the sample is 52 years. A total of 6005 patients were discharged, 20 patients were sent to outpatient clinic, 243 patients left the emergency ward before concluding the exams, 30 patients refused hospitalization, and two patients were transferred to another hospital. The most frequent symptoms that determined admission were the following: abdominal pain (18.9%; n = 1191).; non-specific chest pain (18.7%; n = 1175); lower back pain (12.3%; n = 775); headache (9%; n = 571).

### 3.2. Evolution of the Diagnostic Criteria from Somatoform Disorder (DSM-IV-TR) to SSD (DSM-5)

A total of 32 patients (19 women and 13 men) of the total sample who did not receive a diagnosis of somatoform disorder, fulfill the diagnostic criteria of DSM-5 Somatic Symptoms Disorder, based on the discharge letter. A total of 6 patients had a psychiatric consultation during hospitalization.

A total of 16 patients had a previous psychiatric diagnosis (Anxiety Disorder n = 10; Depressive Disorder n = 4; Substance Abuse (n = 1); Anxiety Disorder/Eating Disorder n = 1), seven patients received a psychiatric diagnosis upon discharge (Anxiety Disorder n = 4; Depressive Disorder n = 2; Personality Disorder n = 1) and nine patients had no previous psychiatric diagnosis and they did not receive a psychiatric diagnosis upon discharge. 

### 3.3. Psychopathological Correlates of SSD Diagnosis 

Chi-square tests showed that the distribution of somatic symptom disorder diagnoses was not significantly different in any of the two genders (χ^2^(1) = 0.31; *p* = 0.58). Additionally, no differences were found with regard to levels of employment (χ^2^(8) = 5.71; *p* = 0.68) or civil status (χ^2^(4) = 4.38; *p* = 0.36).

Logistic regression models are presented in Table 5.

The table includes Odds Ratios (*OR*), indicating the increase in the probability of occurrence of the SSD diagnosis, and their corresponding Confidence Intervals (*CI*) and *p*-values. *CI*s including the value of 1 indicate no significant relationship. Standard Errors (*SE*) associated with the coefficient and Wald χ^2^ are also reported. The Wald χ^2^ tests the null hypothesis that there is no association: if significant, the probability of occurrence of the SSD diagnosis is significantly associated with the corresponding predictor. As can be seen, the model including medical diagnoses as independent variables indicated that the presence of a neurological disease in medical history was negatively associated with the presence of a diagnosis of somatic symptom disorder (*OR* = 0.34; Wald *χ*^2^(1) = 4.75, *p* = 0.03). However, it has to be noted that the overall model was not significant (*χ*^2^(11) = 17.96; *p* = 0.08; Negelkerke R^2^ = 0.13), meaning that medical diseases did not explain a significant percentage of variance in the dependent variable. Given this, we computed a Phi-correlation coefficient among neurological anamnesis only and somatic symptom disorder diagnosis to further explore this association: the correlation was negative and significant (ϕ = −0.13; *p* = 0.03). 

A logistic regression model including psychiatric diagnoses as independent variables was significant (*χ*^2^(3) = 12.16; *p* < 0.01; Negelkerke R^2^ = 0.10). The model correctly classified 92.7% of participants, and indicated that a personality disorder diagnosis in patients with MUPS was associated with increased probability of having a diagnosis of Somatic Symptoms Disorder (*OR* = 16.18; Wald *χ*^2^(1) = 8.26, *p* < 0.01). A marginally significant positive association (*p* = 0.06) also emerged with anxiety disorder but not with depressive disorder.

### 3.4. Healthcare Management Costs

Table 6 shows the overall estimated cost of hospitalizations for patients with MUPS and the costs divided by the hospital wards. 

The total amount is 475,409.73 € with an average cost per year of 47,540.973 €. The highest costs were observed in medical wards, such as Neurology (328,192.09€) followed by Internal Medicine (147,976.16€). The overall estimated cost of examinations, which include blood tests and instrumental examinations, is 119,926.34 €. The overall estimated cost of hospitalizations in surgical wards is 14,495.14 €.

## 4. Discussion

The study was carried out in a secondary setting. Clinical and diagnostic features of somatoform disorder have been debated by authors over the years, without reaching a consensus on which one could be the best and more useful diagnostic classifications. As in previous studies, MUPS were chosen as the basic diagnostic feature to the first selection of the patients [17,25,27,28]. MUPS still remain the main feature of all the diagnostic labels proposed (official ones and alternative ones), except for Somatic Symptom Disorder (according to DSM-5, APA 2013). This section was introduced in order to change the diagnostic paradigm and facilitate the diagnosis, especially for non-specialists [6,29].

In this study, it emerges that a diagnosis of SSD seems more inclusive than diagnosis of somatoform disorder, with 32 patients (11.7%) fulfilling the diagnostic criteria of SSD, which is more than those who received a diagnosis of somatoform disorders (7.9%). This difference, retrospectively observed, could be partly due to a bias linked to the study design since it was not always possible to deduce the way patients present and perceive their symptoms from the discharge letter.

The present study confirms the gender trend observed in another primary care study [19,21,25] with a high prevalence of females with MUPS. Although this prevalence emerged, no statistically significant correlation between the female gender and SSD was detected. The average age of hospitalized patients with MUPS is 45 years, with a prevalence of married and employed people, contrary to what is observed in the literature [30,31]. This result could be influenced by the lack of almost 20–28% of patients’ information. 

The study highlights a relevant comorbidity of MUPS with other psychiatric disorders (39% in previous medical history and in 16% as a new psychiatric diagnosis). Consistent with previous studies [3,15,23], the most frequently detected disorders were Anxiety Disorder, Depressive Disorder and Substance Abuse.

A psychiatric consultation was requested for 75 admissions in 306 hospitalizations; this result is in line with a previous study in outpatients [32]. The discrepancy between the admissions for medically unexplained symptoms and request of specialist consultation could lead to a misdiagnosis or to a treatment proposal not in line with management guidelines of MUPS [33].

The most prescribed treatments were SSRIs and benzodiazepines. In the literature, it emerged that SSRIs are preferred alone or in combination with antipsychotics [34,35,36]. This result is consistent with what emerged in evidence-based literature. In a recent meta-analysis, it emerged that the new generation of antidepressants have very low-quality evidence regarding their effectiveness, even if their effectiveness is balanced against high rates of adverse effects [3]. No data are available for benzodiazepines, but German guidelines for somatoform disorder discourage the use of anti-anxiety medications, especially in elderly people [37,38,39].

We could not evaluate the eventual efficacy of any type of psychotherapy that presents some evidence of being effective [40,41] because this information was not available in the patients’ discharge letters.

Regarding the data on the wards involved in the presentation of MUPS and the most common symptoms presented by the patients, these data differ from the literature, especially concerning Internal Medicine or Primary Care. For example, Kroenke and Mangelsdorff conducted a longitudinal study on the common symptoms in an internal medical setting, highlighting that the most frequent symptoms were chest pain, fatigue and dizziness [42]. This difference could be due to the large number of neurologic patients in our sample, although if we consider the subgroup of patients referring to the Emergency Ward, lower back pain, non-specific chest pain, headache and abdominal pain formed the most common symptomatology. 

With regard to the correlation between medical anamnesis and SSD, there is no evidence that a history of medical disease is associated with a diagnosis of SSD. In other words, patients with MUPS and a neurological diagnosis in medical history may be less likely to receive a somatic symptom disorder diagnosis compared to patients with MUPS and no neurological diagnosis in anamnesis, although further study is necessary to confirm this datum. It is possible to assume that having already received a diagnostic label of a previous neurological disorder, patients are subsequently not diagnosed with appropriate codification of MUPS [43].

From our analyses, a Personality Disorder diagnosis in patients with MUPS was associated with increased probability of having a diagnosis of Somatic Symptoms Disorder. A marginally significant positive association also emerged with Anxiety Disorder, but not with Depressive Disorder. This interesting result highlights the impact of the previous diagnoses on formulating a diagnosis of SSD in patients presenting MUPS. Further investigations are needed to understand those psychopathological correlations. 

From our cost analysis, the neurology ward had the highest overall healthcare expenditure, including the highest cost for laboratory and instrumental exams. This observation could due to the type of examinations, which are predominantly procedures associated with huge healthcare costs. It is interesting to note that psychiatric hospitalization costs incur higher costs than those related to emergency surgery and infectious disease. This could be due to the long hospitalization durations in psychiatry and because patients in emergency surgery did not receive any surgery after clean investigations. With regard to patients admitted in infectious disease, hospitalizations were shorter than in psychiatry and any medications received were not expensive.

As shown in Table 6, the ratio between costs for MUPS in hospitalized patients and overall costs related to hospitalizations for each ward is higher in Neurology (1.9%) than other specialties. This is in line with the prevalence of clinical presentation, as already described in the text. 

As widely described in the literature, this could be used as a guide to reduce any repetitive investigations and to evaluate the need of a psychiatric consultation early. In fact, psychiatric consultation has been identified as a way to support and implement the diagnostic process in order to reach an earlier person-centered psychiatric intervention, while also evaluating personal resources [44,45,46,47]. 

The present study takes into consideration the costs related to part of the diagnostic process, raising the hypothesis that total healthcare costs for patients with MUPS are even more extensive [43]. As shown in the professional literature, this may only be the tip of the iceberg [25] and it represents the reason why it was not possible to compare our data with healthcare costs derived from previous American and European studies in the professional literature [23,25].

As far as we know, few studies on patients with medically unexplained symptoms admitted to hospital exist in the professional literature. The strengths of the present study consist in the investigation of a large number of patients with MUPS; to study clinical, socio-demographic variables and psychopathological correlations involved in the development of Somatic Symptom Disorder; to provide a financial economic estimate of hospitalization costs of patients with MUPS. 

The study presents some limitations, such as the small sample size from non-medical specialties, limiting the possibility to extend the statistical analyses to the whole sample due to the lack of patients’ personal information. 

Further investigations of this research project could possibly extend the study in other areas, such as General Practice and to extend the research to clinics and outcare patient facilities. 

## Figures and Tables

**Table 1 behavsci-09-00080-t001:** Socio-demographic characteristics of the sample.

	Male		Female	
Age (years ± SD)	47 ± 17.0		44 ± 15.9	
	Number	% (of 82)	Number	% (of 191)
**Gender**	82	30	191	70
**Marital status**				
Married	31	37.8	75	39.3
Single	17	20.7	47	24.6
Divorced	3	3.7	20	10.5
Widowed	1	1.2	8	4.1
Not available	26	31.7	45	23.5
**Occupation**				
Salaried	29	35.3	78	40.8
Retired	25	30.5	16	8.4
Housewife	0	0.0	31	16.2
Unemployed	10	12.2	11	5.6
Student	4	4.9	7	3.9
Invalid	1	1.2	6	3.2
Not available	15	18.3	40	20.9

**Table 2 behavsci-09-00080-t002:** Distribution of patients with Medically Unexplained Physical Symptoms (MUPS) in hospital wards.

Ward	Patients (N)	Hospitalizations (N)	Male/Female ratio	Age (Mean) Range	Prevalence
Medical Wards				18–86	
Neurology	125	144	1/3	44.0	3.87%
Internal Medicine	60	61	1/5	49.0	-
Short Unit Stay	51	53	2/3	49.0	0.98%
Infectious Disease	7	7	2/5	50.0	-
Surgical Wards				19–71	
Emergency Surgery	10	10	1/1	42.1	0.96%
Orthopedics	2	2	1/0	36.5	0.02%
Psychiatry	12	14	3/10	49.5 22–67	0.42% -
Audiovestibology	14	15	2/3	43.5 18–60	1.51%

**Table 3 behavsci-09-00080-t003:** Psychiatric comorbidity in patients with MUPS.

(1) Previous Diagnosis			
-Anxiety Disorder	Male	13	25.5%
	Female	38	74.5%
-Depressive Disorder	Male	3	17.6%
	Female	14	82.4%
-Substance Abuse	Male	3	100.0%
	Female	0	0.0%
-Somatoform Disorder	Male	2	70.0%
	Female	1	30.0%
-Personality Disorder	Male	0	0.0%
	Female	2	100.0%
**-Comorbidity**			
AD^1^; PD^2^; SFD^3^	Male	0	0.0%
	Female	3	100.0%
AD^1^; SFD^3^	Male	0	0.0%
	Female	2	100.0%
ED^4^; PD^2^; SFD^3^; AD^1^	Male	1	100.0%
	Female	0	0.0%
AD^1^; DD^6^; SFD^3^	Male	0	0.0%
	Female	1	100.0%
PD^2^; SFD^3^	Male	0	0.0%
	Female	1	100.0%
ED^4^, SFD^3^, AD^1^, SA^4^	Male	0	0.0%
	Female	1	100.0%
ED^4^; AD^1^; SFD^3^	Male	0	0.0%
	Female	1	100.0%
ED^4^; PD^2^; SFD^3^; AD^1^	Male	0	0.0%
	Female	1	100.0%
Others	Male	1	9.1%
	Female	10	90.9%
**-Other**			
Post-Traumatic Stress Disorder	Male	0	0.0%
	Female	2	100.0%
Parasuicide	Male	0	0.0%
	Female	1	100.0%
**(2) Discharge diagnosis**			
-Depressive Disorder	Male	9	53.0%
	Female	8	47.0%
-Anxiety Disorder	Male	3	18.8%
	Female	13	81.2%
-Somatoform Disorder	Male	2	28.5%
	Female	5	71.5%
-Substance Abuse	Male	0	0.0%
	Female	2	100.0%
-Personality Disorder	Male	1	50.0%
	Female	1	50.0%
**-Comorbidity**			
AD^1^; SFD^3^	Male	1	100%
	Female	0	0.0%
DD^6^; SFD^3^	Male	0	0.0%
	Female	1	100.0%
**(3) No diagnosis**			
	Male	43	34.1%
	Female	83	65.8%

AD^1^: Anxiety Disorder; PD^2^: Personality Disorder; SFD^3^: Somatoform Disorder; ED^4^: Eating Disorder; SA^5^: Substance Abuse; DD^6^: Depressive Disorder.

**Table 4 behavsci-09-00080-t004:** Symptoms per unit.

Hospital Ward	Symptoms
Short Stay Unit	Syncope (N = 13); Pain (N = 16); Paraesthesia (N = 5); Headache (N = 9); Vertigo (N = 4); Motor deficit (N = 2); Neurological dysfunction (N = 1); Postural instability + Loss of consciousness (N = 1); Fainting (N = 1); Aphasia (N = 1); Headache + Paraesthesia (N = 1)
Neurology	Headache (N = 50); Paraesthesia (N = 19); Pain (N = 8); Neurological dysfunction (N = 14); Motor deficit (N = 9); Loss of consciousness (N = 4); Motor deficit + Paraesthesia (N = 2); Headache + Vertigo (N = 3); Fainting (N = 2); Headache + Paraesthesia (N = 4); Vertigo (N = 2); Aphasia (N = 1); Headache + Pain (N = 1); Headache + Motor deficit (N = 1); Headache + Fainting (N = 1); Dysphagia (N = 1); Fibromyalgia (N = 1)Postural instability (N = 1); Hypochondria (N = 1); General malaise (N = 1); Blurring (N = 1); Paresis (N = 1); Loss of consciousness + Pain + Paraesthesia (N = 1)
Infectious Disease	Pain (N = 4); Urinary disorders (N = 1); Enteritis (N = 1); Fever (N = 1)
Internal Medicine	Pain (N = 9); Syncope (N = 5); Headache (N = 5) Fainting (N = 3); Paraesthesia (N = 3); Vertigo (N = 3); Fever (N = 3); Vomit (N = 2); Absence (N = 2); Dyspnoea (N = 2); Asthenia (N = 2); Asthenia + Vertigo + Fainting (N = 1); Pain + Impotence (N = 1); Weight loss + Night sweats (N = 1); Headache + Pain (N = 1); Haemorrhage (N = 1); Headache + Aphasia (N = 1); Fainting + Paraesthesia (N = 1); Pain + Nausea (N = 1); Fainting + hypokalaemia (N = 1); Vertigo + Nausea (N = 1); Pain + Nausea (N = 1); Chest tightness (N = 1); Pain + Nausea + Haemorrhage (N = 1); Weight loss (N = 1); Blood pressure increase + Palpitation (N = 1); Dysphagia (N = 1); Syncope + Headache (N = 1); Drowsiness (N = 1); Agitation (N = 1); Tremor (N = 1); Vertigo + Malaise (N = 1);
Emergency Surgery	Pain (N = 8); Pain + Fever (N = 1); Headache + Paraesthesia (N = 1)
Orthopedics	Pain (N = 2)
Psychiatry	Agitation (N = 5); Anxiety (N = 4); Syncope (N = 2); Paraesthesia (N = 1); General malaise (N = 1); Cognitive impairment (N = 1)
Audiovestibology	Hypoacusis (N = 6); Vertigo (N = 5); Vertigo + Pain (N = 1); Chronic Dizziness (N = 1); Postural instability (N = 1); Fainting (N = 1); Fainting + Vertigo (N = 1);

**Table 5 behavsci-09-00080-t005:** Multiple logistic regression predicting the probability of receiving a Somatic Symptoms Disorder (SSD) diagnosis from medical and psychiatric diseases.

	*OR*	*CI for OR*	*SE* ^1^	Wald *χ*^2^ (df = 1)
MEDICAL DISEASE ^2^				
Fibromyalgia	0.39	[0.05, 3.35]	1.09	0.72
Previous medical history	1.20	[0.52, 2.79]	0.42	0.18
Neurological disorders	0.34	[0.13, 0.90]	0.50	4.75*
Neoplasms	1.04	[0.26, 4.13]	0.70	0.00
Metabolic disorders	2.11	[0.76, 5.91]	0.52	2.05
Autoimmune diseases	0.70	[0.16, 3.06]	0.75	0.22
Endocrine diseases	2.41	[0.62, 9.31]	0.69	1.63
Infectious diseases	1.83	[0.41, 8.13]	0.76	0.63
Medical diseases	2.10	[0.86, 5.13]	0.46	2.64
Surgical diseases	0.41	[0.14, 1.21]	0.55	2.62
Accident	1.44	[0.42, 4.88]	0.62	0.34
PSYCHIATRIC DISEASE ^3^				
Depressive Disorder	1.54	[0.40, 5.95]	0.69	0.39
Anxiety Disorder	2.43	[0.94, 6.26]	0.48	3.39
Personality Disorder	16.18 *	[2.42, 108.03]	0.97	8.26 *

^1^*SE* = Standard Error; *OR* = Odds Ratio [Exp(B)]; N = 273; * *p* < 0.05; ^2^ R^2^ = 0.13; omnibus *χ*^2^(11) = 17.96; *p* = 0.08. ^3^ R^2^ = 0.10; omnibus *χ*^2^(3) = 12.16; *p* < 0.01.

**Table 6 behavsci-09-00080-t006:** Costs of hospitalizations (Euro).

			MUPS			Overall	%
Ward	Hosp	ALOH (days)	Total	Each	Exams	Total	MUPS/ Overall
Neurology	144	9	328,192.09	2,263.4	71,441.89	17,474,510	1.9%
Internal Medicine	61	8	147,976.16	2425.8	13,704	43,509,770	0.3%
Short Stay Unit	53	4	71,853.8	1335.72	10,383.95	6,529,696	1.1%
Infectious Disease	7	8	13,482.65	1926.1	9782.80	31,259,630	0.04%
Emergency Surgery	10	5	12,393.54	1652.47	2375.58	12,403,892	0.09%
Orthopedics	2	3	2101.6	1050.82	791.86	76,801,450	0.002%
Psychiatry	14	8	34,129.61	2437.83	5541.03	11,539,717	0.3%
Audiovestibology	15	7	9965.88	664.72	5905.23	6,648,790	0.1%

Hosp: Hospitalization; ALOH: Average length of hospitalization.

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
