# Peer review of "Hospitalized Patients with Medically Unexplained Physical Symptoms: Clinical Context and Economic Costs of Healthcare Management"

_behavsci, 2019, doi:10.3390/bs9070080_

Round 1
Reviewer 1 Report
The topic is very interesting, and I consider it would have a deep impact in public health cost. I would recommend simplifying the biostatistics results and using ORs with their correspondent CI or p-values depending on what the authors want to convey to the reader, would make the lecture more fluid and understandable.
I also recommend to review the study design, I consider this is a cross-sectional study. I made a comment about it in the text.

Author Response
Reviewer 1
The topic is very interesting, and I consider it would have a deep impact in public health cost. I would recommend simplifying the biostatistics results and using ORs with their correspondent CI or p-values depending on what the authors want to convey to the reader, would make the lecture more fluid and understandable.
We thank the reviewer for this comment and for the corresponding comments within the text. As suggested, we included CIs in Table 4 and we added all exact p-values for the significant and marginally significant associations within the text. Also, columns marked with the sign “*” in the table indicate a p-value lower than .05 (statistically significant).
The reviewer’s comment also gives us the chance to be more clear about the statistics we reported: the Exp(B) reported in the table is indeed the Odds Ratio, as it is the exponentiation of the B coefficient expressed in log-units (not reported here, as it is not easily interpretable): an OR of 1 indicates no relationship, an OR greater than 1 indicate a positive relationship, while an OR lower than 1 indicates a negative relationship. Therefore, a confidence interval including the value of 1 indicates a non-significant association. We briefly explained this in the Results section. Moreover, for the sake of clarity, we removed the header “Exp(B)” from the table, as well as any reference to Exp(B) within the text, and substituted “Exp(B)” with “OR” (Odds Ratio), following the reviewer’s suggestion.
I also recommend to review the study design, I consider this is a cross-sectional study. I made a comment about it in the text.
We reviewed the study design, considering the research a cross-sectional study, as suggested. We also provide responses to reviewer’s comments in the text and below:
- Make a short description of the acronyms.
- A short description of the acronym was added, as suggested.
- “largely cross-sectional”
- We modified the sentence in the text.
- I consider that this is a cross-sectional study because the authors are analyzing data from a determined period of time, it is a snapshot of the diagnosis prevalence and its probable association with other pathologies. It isn´t a cohort study because there is no cohort that is exposed to certain risk factor and followed (prospective or retrospective) along the time to determine if they developed a disease.
- We reviewed the design of the study, considering the research as a cross-sectional study.
- This is not necessary to state in the text
- We cut out the sentence, as suggested.
- “are”
- We have changed the word “are” with “were”, as suggested.
- I recommend to use ORs instead of Exp(B), it would be easier for the reader to understand that association measure. I also consider that it would be useful to state the correspondent confidence intervals or p-values, which could help the readers to understand if they are statistically significant or not.
- The Exp(B) reported in the table is indeed the Odds Ratio, as it is the exponentiation of the B coefficient expressed in log-units (not reported here, as it is not easily interpretable): an OR of 1 indicates no relationship, an OR greater than 1 indicate a positive relationship, while an OR lower than 1 indicates a negative relationship. Therefore, a confidence interval including the value of 1 indicates a non-significant association. We briefly explained this in the Results section. Moreover, we removed the header “Exp(B)” from the table, as well as any reference to Exp(B) within the text, and substituted “Exp(B)” with “OR” (Odds Ratio), following the reviewer’s suggestion.
- I recommend to state something very brief about the SE and Wald chi-square in the text. This information means more about the fit of the model than about the association of the independent and dependent variables. Then, I consider that CI or p-values would be more intuitive for the readers.
- Both the SE and Wald chi-square are associated with the B regression coefficient expressed in log-units (not reported here); however, they allow the computation of p-values testing the null hypothesis that no associations exists among the variables. As suggested, we briefly mentioned the SE and Wald chi-square within the text, explaining their utility. As mentioned before, CIs are now reported in Table 4, whereas significant or marginally significant p-values associated with the Wald chi-square are reported in the text.
- OR and CI? pr p-value would be very useful.
- As suggested, we included CIs in Table 4 and we added all exact p-values for the significant and marginally significant associations within the text. Also, columns marked with the sign “*” in the table indicate a p-value lower than .05 (statistically significant).
- It would be important to state the currency in the title.
- We stated the currency in the title.
- Please clarify this paragraph.
- We clarified the paragraph changing the statement.
- Use the word had instead of associated because there was not a logistic regression to determine such relationship
- We changed the sentence using the word "had" instead of "associated", as suggested.

Reviewer 2 Report
This manuscript aims at describing the clinical and socio-demographic profile of patients with medically unexplained physical symptoms (MUPS), explore the correlates of Somatic Symptom Disorder (SSD) diagnosis, and to estimate economic costs related to hospital management of MUPS. The study involved 273 patients showing MUPS and the authors reported on the number of socio-demographic factors and correlates.
Given the prevalence of MUPS and the recent changes in DSM-5, research that expands our understanding of SSD is of high importance. However, the manuscript could benefit from a number of modifications. In particular, the prevalence of MUPS per unit should be reported, because just mentioning the number of MUPS per unit does not give readers full information about the severity of the MUPS problem (e.g., it could be related to the general number of patients per unit). Second, the clarity of the results section should be improved, because in the current format is it difficult to follow. Comments below are intended to further strengthen the paper.
Abstract
“worker patients” I guess it should be changed to employed patients (also L281)
Introduction
L39-40. This sentence does not have a verb.
L48-49. It is not clear what the authors mean with “patients suffering from chronic medical conditions can also be diagnosed and receive treatment [2].”
I appreciate that the authors explain the advantages and difficulties related to the SSD classification. However, because the authors shift between SSD and MUPS, it would be important to be precise in the statements such as L55 “based on DSM criteria DSM criteria resulted in low prevalence of this disease”- of SSD or somatoform disorder or MUPS?
In addition, I was a bit surprised seeing the low prevalence rates of somatoform disorders. For example, de Waal et al (2004) reported the prevalence of 16.1% among patients of general practitioners.
L61 – abbreviation SSI should be explained
L63-64. Incorrect reference #16. “Up to one third of all people presenting with physical
symptoms have MUPS [15,16]” Reference 16 refers to a case report, not a prevalence study.
L76. The authors could add the reference to a systematic review by Konnopka et al (2012) on the economics of MUS.
Results
This section is rather difficult to follow. The authors could consider adding subheaders to improve readability.
Table 1. For the subsample description, I would find it more informative to see the proportions within each sample instead of the percentage of the total sample (e.g. married males: 37.8 % (and not 11.4%); married females 39.2% (and not 27.4%)). This would allow the reader to compare those two subsamples with greater ease.
Table 2. The clarity of this table could be improved.
- The subcategories should be more clearly indicated (now it looks as if Psychiatry is under “Surgical wards”.
- What is represented in the column M/F? The age range is not mentioned for all categories, and for “emergency” no mean age is provided.
- The number of patients and admissions in the emergency ward is very large (5735/6921), but is not included in the statement “the overall hospitalization detected were 306”. Also, if the inclusion criterium was age above 18, why is age range between 10 and 101? As the authors discuss this sample fully in the text, maybe this part could be removed from the table to decrease the confusion.
- To improve clarity, the wording should be consistent with the main text:
o “Audiovestibology” should be replaced with otorhinolaryngology
o Hospitalizations/admissions
I would be really interested to see the prevalence of MUPS per unit, i.e. the ratio of MUPS patients to all hospitalized patients per unit. This would be an interesting and important addition to the literature.
Table 3. The structure of the table should be improved because right now it is difficult to evaluate the content of this table. Again, the percentages should refer to the subsample and not to the total sample.
L196-199. Are those numbers based on what is reported in the table? Because there is a discrepancy for both males and females, with n’s based on the table: 24 and 76, respectively.
L200-202.
L206-208. This is not mentioned in the table. How many patients overall (including the ones with somatoform disorder diagnosis) fulfilled the diagnostic criteria for SSD? This could say more about the overlap between the constructs.
L209-2013. Does this paragraph refer to the SSD patients? If those 2 paragraphs (this and the previous one) refer to SSD, maybe they should receive a separate subheader?
L220-221. Could the authors write a bit more about the symptomatology per unit?
3.1. and 3.2 have exactly the same header.
The sample size of 32 is quite low for the logistic regressions with so many predictors. Could this lead a lack of significant outcomes?
Table 5. In the discussion section, the authors refer to the number of hospitalizations and the average length of the hospitalizations per unit. This information should be included in the table. Maybe information about the average cost of hospitalization (per unit, including all patients, not only MUPS) should be included. This would allow for a better understanding of the economic costs related to MUPS relative to the average cost of hospitalization.
Discussion
L266-267. “Moreover, previous studies have been conducted consisted with this study design’s characteristic [17, 23, 25].” This sentence is unclear.
L270. What does “all” refer to?
L275. The percentage of patients with somatoform disorder could be added to make the comparison easier.
L284-287. Does this refer to SSD or to MUPS?
Language typos
L55 “who”
L57 “don’t”
L66 “between”
L98 “letters”
References:
de Waal, M. W. M., Arnold, I. A., Eekhof, J. A. H., & van Hemert, A. M. (2004). Somatoform disorders in general practice: Prevalence, functional impairment and comorbidity with anxiety and depressive disorders. The British Journal of Psychiatry, 184, 470–476. https://doi.org/10.1192/bjp.184.6.470
Konnopka, A., Schaefert, R., Heinrich, S., Kaufmann, C., Luppa, M., Herzog, W., & König, H.-H. (2012). Economics of medically unexplained symptoms: A systematic review of the literature. Psychotherapy and Psychosomatics, 81, 265–275. https://doi.org/10.1159/000337349
Author Response
Reviewer 2
This manuscript aims at describing the clinical and socio-demographic profile of patients with medically unexplained physical symptoms (MUPS), explore the correlates of Somatic Symptom Disorder (SSD) diagnosis, and to estimate economic costs related to hospital management of MUPS. The study involved 273 patients showing MUPS and the authors reported on the number of socio-demographic factors and correlates.
Thank you for the reviewer’s revision of our paper. We are sure that the suggestions can greatly improve the quality of the paper.
Given the prevalence of MUPS and the recent changes in DSM-5, research that expands our understanding of SSD is of high importance. However, the manuscript could benefit from a number of modifications. In particular, the prevalence of MUPS per unit should be reported, because just mentioning the number of MUPS per unit does not give readers full information about the severity of the MUPS problem (e.g., it could be related to the general number of patients per unit). Second, the clarity of the results section should be improved, because in the current format is it difficult to follow. Comments below are intended to further strengthen the paper.
We reported the prevalence of MUPS for all units whose data were available from hospital electronic registers. Prevalence of MUPS are shown in table 2. We improved the clarity of the results section, as the reviewer suggested.
Abstract
“worker patients” I guess it should be changed to employed patients (also L281)
We changed “worker patients” with “employed patients, as suggested.
Introduction
L39-40. This sentence does not have a verb.
We corrected the sentence properly.
L48-49. It is not clear what the authors mean with “patients suffering from chronic medical conditions can also be diagnosed and receive treatment [2].”
We clarified the meaning of the sentence.
I appreciate that the authors explain the advantages and difficulties related to the SSD classification. However, because the authors shift between SSD and MUPS, it would be important to be precise in the statements such as L55 “based on DSM criteria DSM criteria resulted in low prevalence of this disease”- of SSD or somatoform disorder or MUPS?
We pointed out that the subject of the sentence was somatoform disorder.
In addition, I was a bit surprised seeing the low prevalence rates of somatoform disorders. For example, de Waal et al (2004) reported the prevalence of 16.1% among patients of general practitioners.
The percentage 0.4% refers to a systematic review from Creed and Barsky (2003), and it represents the median rate of 10 studies. However, we considered and cited the recommended paper by Waal et al. (2004), which, despite being a single study, it involves a very large population.
L61 – abbreviation SSI should be explained
The abbreviation “SSI” has been explained in the text, as suggested.
L63-64. Incorrect reference #16. “Up to one third of all people presenting with physical symptoms have MUPS [15,16]” Reference 16 refers to a case report, not a prevalence study.
We corrected the references properly.
L76. The authors could add the reference to a systematic review by Konnopka et al (2012) on the economics of MUS.
We added the reference as suggested.
Results
This section is rather difficult to follow. The authors could consider adding subheaders to improve readability.
We added subheaders to improve the readability of the section.
Table 1. For the subsample description, I would find it more informative to see the proportions within each sample instead of the percentage of the total sample (e.g. married males: 37.8 % (and not 11.4%); married females 39.2% (and not 27.4%)). This would allow the reader to compare those two subsamples with greater ease.
We modified table 1 reporting the proportion of each sample.
Table 2. The clarity of this table could be improved.
We improved the clarity of the table.
- The subcategories should be more clearly indicated (now it looks as if Psychiatry is under “Surgical wards”.
- We indicated the subcategories in Table 2 in a clearer way.
- What is represented in the column M/F? The age range is not mentioned for all categories, and for “emergency” no mean age is provided.
- The column M/F represents the male:female sex ratio; we specified it better in the table. We reported the mean age for “emergency surgery”.
- The number of patients and admissions in the emergency ward is very large (5735/6921), but is not included in the statement “the overall hospitalization detected were 306”. Also, if the inclusion criterium was age above 18, why is age range between 10 and 101? As the authors discuss this sample fully in the text, maybe this part could be removed from the table to decrease the confusion.
- We removed “Emergency ward” part from the table, as suggest.
- To improve clarity, the wording should be consistent with the main text:
· “Audiovestibology” should be replaced with otorhinolaryngology
· We reported “Audiovestibology” instead of “Otorhinolaryngology”, because in the hospital involved in the research these units are considered as two different wards.
· Hospitalizations/admissions
· We changed “hospitalizations” with “admissions”, as suggested.
I would be really interested to see the prevalence of MUPS per unit, i.e. the ratio of MUPS patients to all hospitalized patients per unit. This would be an interesting and important addition to the literature.
As already stated, we reported the prevalence of MUPS for all units whose data were available from hospital electronic registers. Prevalence of MUPS are shown in table 2
Table 3. The structure of the table should be improved because right now it is difficult to evaluate the content of this table. Again, the percentages should refer to the subsample and not to the total sample.
We improved the structure of table 3 reporting the percentages of the subsamples, as suggested.
L196-199. Are those numbers based on what is reported in the table? Because there is a discrepancy for both males and females, with n’s based on the table: 24 and 76, respectively.
Thank you for your observations. We revised correctly the numbers in tables and also in the text.
L200-202.
L206-208. This is not mentioned in the table. How many patients overall (including the ones with somatoform disorder diagnosis) fulfilled the diagnostic criteria for SSD? This could say more about the overlap between the constructs.
We clarified this point in the text.
L209-2013. Does this paragraph refer to the SSD patients? If those 2 paragraphs (this and the previous one) refer to SSD, maybe they should receive a separate subheader?
We separated this paragraph from the text more correctly, and we specified for it a separate subheader.
L220-221. Could the authors write a bit more about the symptomatology per unit?
We reported a table (Table in which symptoms per unit are shown.
3.1. and 3.2 have exactly the same header.
We modified the headers properly.
The sample size of 32 is quite low for the logistic regressions with so many predictors. Could this lead a lack of significant outcomes?
Both logistic regressions were conducted on a sample of 273 patients. We thank the reviewer for these comments, as it gave us the possibility to clarify this information. We added the sample size at the end of the table, within the table captions.
Table 5. In the discussion section, the authors refer to the number of hospitalizations and the average length of the hospitalizations per unit. This information should be included in the table. Maybe information about the average cost of hospitalization (per unit, including all patients, not only MUPS) should be included. This would allow for a better understanding of the economic costs related to MUPS relative to the average cost of hospitalization.
Information about the length of hospitalizations and the average cost of hospitalization per unit have been included in the present table (now Table 6)
Discussion
L266-267. “Moreover, previous studies have been conducted consisted with this study design’s characteristic [17, 23, 25].” This sentence is unclear.
We clarified the sentence in the text, as suggested.
L270. What does “all” refer to?
We explained better in the text.
L275. The percentage of patients with somatoform disorder could be added to make the comparison easier.
We modified the sentence reporting the percentages of patients who received the diagnosis of somatoform disorder in comparison with patients fulfilling the diagnostic criteria for SSD.
L284-287. Does this refer to SSD or to MUPS?
It refers to MUPS.
Language typos
L55 “who”
L57 “don’t”
L66 “between”
L98 “letters”
We corrected the typos errors.
References:
de Waal, M. W. M., Arnold, I. A., Eekhof, J. A. H., & van Hemert, A. M. (2004). Somatoform disorders in general practice: Prevalence, functional impairment and comorbidity with anxiety and depressive disorders. The British Journal of Psychiatry, 184, 470–476. https://doi.org/10.1192/bjp.184.6.470
Konnopka, A., Schaefert, R., Heinrich, S., Kaufmann, C., Luppa, M., Herzog, W., & König, H.-H. (2012). Economics of medically unexplained symptoms: A systematic review of the literature. Psychotherapy and Psychosomatics, 81, 265–275. https://doi.org/10.1159/000337349
We added these citations in the reference section.

Round 2
Reviewer 2 Report
The authors have improved the manuscript, congratulations! I have only a few remaining comments regarding percentages reported in the tables. As many readers focus only on the percentages and not the actual numbers, it is important that those numbers are clear. I am adding my suggestions in a separate file.

Author Response
Reviewer 1
Table 1
Thank you for your comments. We modified table 1, as suggested.
Table 2
We corrected the error regarding hyphen.
We corrected the number after digits for average age.
Table 3
We modified the table properly, as suggested.
L192: n=101
We corrected the error, as suggested.
